# Protective Abilities of an Inhaled DPI Formulation Based on Sodium Hyaluronate against Environmental Hazards Targeting the Upper Respiratory Tract

**DOI:** 10.3390/pharmaceutics14071323

**Published:** 2022-06-22

**Authors:** Juhura G. Almazi, Dina M. Silva, Valentina Trotta, Walter Fiore, Hui X. Ong, Daniela Traini

**Affiliations:** 1Respiratory Technology, Woolcock Institute of Medical Research, Sydney, NSW 2037, Australia; juhura.almazi@sydney.edu.au; 2Ab Initio Pharma Pte Ltd., 63-73 Missenden Road, Camperdown, NSW 2050, Australia; dina.silva@ab-initio-pharma.com; 3HollyCon Italy Pte Ltd., Srl, Via Danimarca 21, 20083 Gaggiano, Italy; v.trotta@hollyconitaly.com; 4SOFAR Spa, Via Firenze, 40, 20060 Trezzano Rosa, Italy; walter.fiore@sofarfarm.it; 5Macquarie Medical School, Department of Biological Sciences, Faculty of Medicine, Health and Human Sciences, Macquarie University, Macquarie Park, NSW 2109, Australia

**Keywords:** dry powder inhaler, barrier, protection, air-liquid interface, sodium hyaluronate

## Abstract

The exposure of lung epithelium to environmental hazards is linked to several chronic respiratory diseases. We assessed the ability of an inhaled dry powder (DPI) medical device product (PolmonYDEFENCE/DYFESA^TM^, SOFAR SpA, Trezzano Rosa, Italy), using a formulation of sodium hyaluronate (Na-Hya) as the key ingredient as a defensive barrier to protect the upper respiratory tract. Specifically, it was evaluated if the presence of the barrier formed by sodium hyaluronate present on the cells, reducing direct contact of the urban dust (UD) with the surface of cells can protect them in an indirect manner by the inflammatory and oxidative process started in the presence of the UD. Cytotoxicity and the protection capability against the oxidative stress of the product were tested in vitro using Calu-3 cells exposure to UD as a trigger for oxidative stress. Inflammation and wound healing were assessed using an air-liquid interface (ALI) culture model of the Calu-3 cells. Deposition studies of the formulation were conducted using a modified Anderson cascade impactor (ACI) and the monodose PillHaler^®^ dry powder inhaler (DPI) device, Na-Hya was detected and quantified using high-performance-liquid-chromatography (HPLC). Solubilised PolmonYDEFENCE/DYFESA^TM^ gives protection against oxidative stress in Calu-3 cells in the short term (2 h) without any cytotoxic effects. ALI culture experiments, testing the barrier-forming (non-solubilised) capabilities of PolmonYDEFENCE/DYFESA^TM^, showed that the barrier layer reduced inflammation triggered by UD and the time for wound closure compared to Na-Hya alone. Deposition experiments using the ACI and the PillHaler^®^ DPI device showed that the majority of the product was deposited in the upper part of the respiratory tract. Finally, the protective effect of the product was efficacious for up to 24 h without affecting mucus production. We demonstrated the potential of PolmonYDEFENCE/DYFESA^TM^ as a preventative barrier against UD, which may aid in protecting the upper respiratory tract against environmental hazards and help with chronic respiratory diseases.

## 1. Introduction

The airway epithelium is directly exposed to the external environment and, consequently, pollutants, such as cigarette smoke, dust, silica, and other biological threats that can result in infection and inflammation. This, in combination with individual genetic and lifestyle factors, can lead to susceptibility to developing chronic bronchopulmonary diseases such as asthma, chronic obstructive pulmonary disease (COPD), cystic fibrosis and acute respiratory distress syndrome. Biochemical signalling for inflammation and oxidative stress are the drivers for a number of these lung diseases [1,2,3,4]. Different research studies present in the literature are focused on the prevention of chronic bronchopulmonary diseases by the modulation and reduction of inflammation and oxidative stress. While in the present research work, we verify whether the presence of one mechanical barrier layer on the cells could reduce the possibility of starting the inflammatory and oxidating mechanism caused by the direct contact with external environmental hazards, such as UD, which is produced in urbanised areas because of environmental pollution.

Specifically, hyaluronic acid, a major component of the extracellular matrix and is composed of glucuronic acid and N-acetyl glucosamine units, has recently become the focus of respiratory-related studies. The central role of hyaluronic acid in the respiratory system has been associated with tissue homeostasis through different factors such as hydric balance and biomechanical integrity [5,6,7,8]. Characteristically, hyaluronic acid or sodium hyaluronate is known to be an ingredient that can form a physical barrier (hydrogel) when exposed to aqueous solutions that might play an important role in the biomechanical integrity of the tissues [9,10]. Sittek et al. showed when dissolved in artificial saliva, the presence of sodium hyaluronate increased the viscosity of the saliva. The polymers absorb the saliva forming three-dimensional solution structures stabilised by hydrogen or ionic bonds, thus creating a physical barrier [11]. Hyaluronic acid is very often used in combination with other compounds such as mannitol. Mannitol is a naturally occurring sugar alcohol found in the sap of the manna tree and many vegetables and is generally regarded as safe. It can be used as an excipient in dry powder inhaler (DPI) formulations, where it can act as a carrier molecule to enhance the flow ability and inhalation performance of the product [12,13]. Moreover, mannitol has been theorised to help preserve the viscoelastic properties of hyaluronic acid [14].

PolmonYDEFENCE/DYFESA^TM^ is a brand name of a medical device product that consists of a DPI formulation based on sodium hyaluronate and mannitol as principal ingredient and excipient, respectively, delivered with the disposable monodose PillHaler^®^ DPI device. The associated formulation was developed as a preventative therapeutic, which can act as a defensive barrier to protect the upper respiratory tract from environmental hazards.

With the present study, we hypothesised that the formation of the barrier layer by PolmonYDEFENCE/DYFESA^TM^ blend on the epithelium will exhibit greater protective effects against environmental hazards in Calu-3 epithelia in vitro, resulting in a reduction of inflammation and oxidative stress in the epithelium through an indirect protective mechanism based on the barrier layer. This is because the presence of the barrier layer prevents the environmental hazard from being in direct contact with the epithelium.

To test this hypothesis, our study aimed to conduct a comprehensive assessment (Appendix A) of the protective effects in the upper respiratory tract and the primary bronchi of the barrier-forming product PolmonYDEFENCE/DYFESA^TM^ on the lung epithelium in vitro. This will be achieved by, firstly, determining its protective abilities against environmental hazards such as exposure to environmental pollutants and wounds. Secondly, by establishing the deposition of sodium hyaluronate delivered using the PillHaler^®^ DPI device into a pharmacopoeia-approved impactor (Anderson cascade impactor).

## 2. Materials and Methods

### 2.1. Materials

Hyaluronic acid (Altergon Italia Srl, Morra De Sanctis, Italy), mannitol (Giusto Faravelli S.p.A., Milan, Italy), the PolmonYDEFENCE/DYFESA^TM^ (medical device product formulated and patented by SOFAR S.p.A., Trezzano Rosa, Milan, Italy) and the PillHaler^®^ DPI Device patented by Hollycon Italy (Gaggiano, Milan, Italy) and used as supplied. Urban dust (UD) was purchased from the National Institute of Standards and Technology (NIST, Gaithersburg, MD, USA). Calu-3 cells were supplied by American Type Cell Culture Collection (ATTC, Rockville, MD, USA). Transwell^®^ polyester cell inserts (0.33 cm^2^ surface area, 0.4 μm pore size) and Snapwell™ polyester cell inserts (1.12 cm^2^ surface area, 0.4 μm pore size) were obtained from Corning by Sigma-Aldrich (Sydney, Australia). Triton^®^ X-100, L-ascorbic acid, 2′,7′-dichlorofluorescein diacetate (DCFH-DA), Menadione and 200 mM L-glutamine solution were purchased from Sigma-Aldrich (Milan, Italy, and Sydney, Australia). Other cell culture reagents, including Dulbecco’s Modified Eagle’s medium/F-12, phosphate buffer saline (PBS) and foetal bovine serum (FBS), were obtained from Gibco by ThermoFisher Scientific (Sydney, Australia). Methyl tetrazolium salt (MTS) reagents were purchased from Promega (Sydney, Australia). Enzyme-linked immunoassay (ELISA) kits for the determination of the inflammation markers interleukin-6 (IL-6) and interleukin-8 (IL-8) were obtained from BD Bioscience (Sydney, Australia). Water was purified by Milli-Q reverse Osmosis (Molsheim, France).

### 2.2. Cell Culture

The lung epithelial cancer-derived Calu-3 cell line (ATCC HTB-55) was chosen as in vitro respiratory epithelium model. Cells were cultured between passages 26 to 36 in 75 cm^2^ flasks containing Dulbecco’s Modified Eagle’s medium/F-12 enriched with 10% (*v*/*v*) foetal bovine serum (FBS), 1% (*v*/*v*) non-essential amino acids solution and 1% (*v*/*v*) L-glutamine solution. Cells were maintained in a humidified 95% air, 5% CO_2_ atmosphere at 37 °C until confluency was reached. The medium was replaced three times a week and cells were passaged according to American Type Culture Collection Recommendations—ATCC guidelines. This cell line has been well optimised to reflect the main characteristics and secretory activity of the airway epithelium when grown at an air-liquid interface (ALI) [15]. For the ALI model, cells were seeded at a density of 7.92 × 10^4^ cells/insert on a Transwell polyester insert (0.33 cm^2^ growth area) containing 100 µL in the apical chamber and 600 µL in the basolateral chamber. The medium from the apical chamber was removed after 24 h from seeding and every day afterwards until an ALI was achieved, while the medium from the basolateral chamber was replaced every second day up to 14 days of culture. Snapwell polyester inserts (1.12 cm^2^ growth area) were used for impaction studies with a cell density of 1.58 × 10^5^ cell/insert and ALI culture was established using 2 mL of media in the basolateral chamber.

### 2.3. Cytotoxicity Assay

MTS assay was performed on Calu-3 cells to measure cytotoxicity on cellular metabolic activity. MTS assay is a colorimetric test based on tetrazolium reduction into formazan. This reaction occurs only in active metabolic cells [16]. Briefly, 100 μL of 5 × 10^4^ cells/well were seeded into a 96 well-plate. After 48h, cells were exposed to pre-warmed media containing a series of 2-fold dilutions of urban dust, the blend, and equivalent concentrations of Na-Hya and mannitol. Background controls (medium) and untreated controls (untreated cells) were included in the experiment, as well as a positive control containing 20% DMSO. A blank plate (cell-free) with equal concentrations of urban dust was used to correct for absorbance reading from the urban dust alone. After 2 h or 24 h of incubation with treatments at 37 °C, in a humidified atmosphere at 5% CO_2_, Calu-3 cells were incubated for another 2 h, in the same conditions, with 20 μL of MTS solution (20% *v*/*v*). Finally, the 96 well-plate was read at 490 nm using a SpectraMax M2 plate reader. The absorbance values were directly proportional to cell viability (%). Experiments were performed in triplicate. Data were expressed as % cell viability relative to untreated control and plotted against compound concentrations (mg/mL).

### 2.4. Analysis of Intracellular Reactive Oxygen Species (ROS)

Oxidative stress was evaluated by quantifying intracellular ROS produced by Calu-3 cells treated with 0–1 mg/mL % Na-Hya and equivalent concentration of the PolmonYDEFENCE/DYFESA^TM^, with and without UD induction. ROS levels were determined by converting the non-fluorescent DCFH-DA into the fluorescent dichlorofluorescein (DCF). Briefly, 100 μL of Calu-3 cells were seeded with a density of 5 × 10^4^ cells/well into a 96 well-plate (black, clear-bottom) and incubated overnight at 37 °C in a humidified atmosphere at 5% CO_2_. Afterwards, Calu-3 were incubated in the dark for 30 min (37 °C, 5% CO_2_) with 100 μL of 5 μM DCFH-DA [17]. Then, the DCFH-DA-containing medium was removed and 100 μL of treatments were added to DCFH-DA-loaded cells, protected from light. Background controls (medium), untreated and unlabelled controls (untreated and unlabelled cells), untreated controls (untreated cells), negative controls (cells treated with 5 mM N-Acetyl Cysteine) and positive controls (cells treated with 100 µM menadione) were included in the experiment. Plates were read immediately at time 0 min and after incubation (37 °C, 5% CO_2_) at the annotated times by a SpectraMax microplate reader with an excitation filter set at 485 nm and an emission filter set at 520 nm. Experiments were performed in triplicate and results were expressed as fold change of ROS production over time 0min. The antioxidant activity of the samples was also investigated by their ability to reduce oxidative stress in Calu-3 cells after induction of ROS production by UD.

### 2.5. Transepithelial Electrical Resistance (TEER)

Transepithelial electrical resistance (TEER) of Calu-3 cells in ALI culture was measured as described previously [18]. Briefly, pre-warmed Hanks’ Balanced Salt Solution (HBSS) was added to the apical chamber and allowed to equilibrate for 30 min at 37 °C under 5% CO_2._ TEER was measured using EVOM2^®^ epithelial voltohmmeter (World Precision Instruments, Sarasota, FL, USA) connected to STX-2 chopstick electrodes at the annotated conditions. Blank controls (cell-free inserts containing HBSS) and untreated controls (inserts of cells in medium) were included in the study. Experiments were performed in triplicate. TEER (Ω cm^2^) was calculated from the measured potential resistance difference (Ω) between the apical and basolateral sides, normalised by subtracting the blank insert and multiplying by the surface area of the Transwell or Snapwell inserts, according to the following equation:*TEER* (Ω cm^2^) = (*Resistance_test_* − *Resistance_blank_*) × *Area*
*of*
*well*
*insert*(1)

### 2.6. Sodium Fluorescein Paracellular Permeability

The functionality of tight junctions and paracellular permeability of the cell layer was investigated using the sodium fluorescein permeability assay. Briefly, sodium fluorescein (2.5 mg/mL) (Sigma Aldrich) was added to the apical chamber and pre-warmed HBSS was added to the basolateral chamber. Transwells or Snapwells were incubated for 4 h at 37 °C with 5% CO_2_, with basolateral samples (100 µL) collected at 0, 0.25, 0.5, 0.75, 1, 1.5, 2, 3 and 4 h to measure the rate of transport (flux) of the sodium fluorescein from the apical chamber to the basolateral chamber. For analysis, the collected basolateral sample fluorescence was measured using the SpectraMax M2 plate reader (excitation: 485 nm; emission: 538 nm). The permeation coefficient (Papp) was calculated according to equation 2, where V is the volume in the basolateral chamber, A is the surface area of the Transwell membrane, $C_{0}$ is the initial concentration in the apical chamber, and dC/dt is flux (cumulative) of Na-Flu through the membrane.
(2)$$Papp=\left(V/AC_{0}\right)\left(dC/dt\right)\;$$

### 2.7. Pro-Inflammatory Marker Expression on ALI Modelled Calu-3 Epithelial Layer

The inflammatory response to exposure to urban dust with and without the PolmonYDEFENCE/DYFESA^TM^ barrier layer was evaluated in the ALI-generated Calu-3 epithelial layers by the detection of the pro-inflammatory cytokines IL-6 and IL-8 using ELISA kits according to the manufacturer’s instructions. The inflammatory trigger UD and test compounds were deposited onto the epithelial layers as a suspension in HPFH (2H,3H-decafluoropentane, used as model propellant), as previously reported [19,20]. HPFH is a hydrophobic and highly volatile propellant, which can be handled as a liquid at ambient pressure and evaporates rapidly after exposure to air, thus leaving the epithelial layer exposed to the dry powders only. After 24 h of exposure, the media from the basolateral chamber was collected for interleukin quantification.

### 2.8. Wound Healing Study with ALI Culture

The wound-healing assay, also known as the scratch assay, was performed on the ALI model of Calu-3 cells. Experiments were performed after 14 days of ALI culture. On the apical side of the cell layer, a scratch was made with a pipette tip (P200 μL) along the diameter of the Transwell membrane. Na-HYA and the blend were deposited on top of the wound using HPFP and control cells were treated with an equal volume of HPFP only. Transwells were kept in a humidified chamber at 37 °C in a 5% CO_2_ atmosphere and 95% humidity, and the wound was observed using a Nikon Eclipse Ti microscope (Nikon, Tokyo, Japan) with Coolsnap ES2 camera. Pictures were taken every 20 min for the first 2 h, followed by every 30 min for 18 h using NIS-Elements (version 3.22.01, Nikon Instruments Inc., New York, NY, USA) after formulation deposition. The images were analysed using Fiji ImageJ, measuring the wound closure area using an in-house macro. The percentage of wound closure was calculated using the following Equation (3), where *A_t_* is the wound area at a given time and *A*_0_ is the initial wound area.
%*Wound*
*Size* = 100 × (*A_t_*/*A*_0_)(3)

### 2.9. Impaction Studies Using the Andersen Cascade Impactor (ACI) 

The deposition profile of the product PolmonYDEFENCE/DYFESA^TM^ across the ACI stages was investigated. The eight-stage ACI, including a USP induction port, was connected to a rotary vein pump (Westech Scientific Instruments, Essex, UK) and the flow rate was adjusted to 60 L/min using a calibrated flow meter (TSI Model Instruments, Shoreview, MN, USA). To minimise particle bouncing, 50 µL of Brij 35: glycerol: ethanol (10:50:40 *v*/*v*/*v*) solution was used to coat ACI plates. After a single shot (dose 30 mg/shot) deposition at 60 L/min for 4 s, the ACI was disassembled, and every stage was washed separately using HPLC grade water to collect the blend from each section. Emitted dose (ED) was defined as the amount of DPI that leaves the device (mouthpiece to the ACI). Experiments were performed in triplicate and Na-Hya was quantified using high-performance liquid chromatography (HPLC).

### 2.10. Sodium Hyaluronate Chemical Quantification by HPLC

Na-Hya detection and quantification were conducted using high-performance liquid chromatography (HPLC) system equipped with SPD-20A UV–Vis detector, an LC-20AT liquid chromatography, a SIL-20A HT autosampler (Shimadzu, Kyoto, Japan) and a BioSep SEC-S2000 column (300 × 7.8 mm, 5 µm, 145A, Phenomenex, Torrance, CA, USA). The mobile phase was 0.05M KH_2_PO_4_, pH 7.0. Samples were analysed at 205 nm, a flow rate of 1 mL/minute and an injection volume of 10 µL. Linearity was obtained between 2.5 and 500 µg/mL (R^2^ = 0.99) with a retention time of 5.0 min.

### 2.11. Scanning Electron Microscopy (SEM)

Powder samples were placed on adhesive black carbon tabs and mounted onto aluminium stubs. The samples were gold coated with a sputter coater (BAL-TEC SCD 005, Tokyo, Japan) and particles were examined under a scanning electron microscope (JEOL-JCM 6000 NeoScope Benchtop SEM, Tokyo, Japan) at 100× magnifications using 15 keV accelerating voltage.

### 2.12. Statistical Analysis

The data are presented as the mean ± standard deviation of three independent experiments. Statistical analysis was performed using Prism software version 8.0 (GraphPad, San Diego, CA, USA). Means were compared by one-way analysis of variance (ANOVA) followed by the annotated tests for multiple comparisons.

## 3. Results and Discussion

### 3.1. Oxidative Stress on Calu-3 Cells Induced by Urban Dust Is Reduced by Co-Incubation with PolmonYDEFENCE/DYFESA^TM^

Environmental pollutants such as urban dust have been shown to induce varying levels of oxidative stress on the lung epithelium [21]. Thus, in this study, a linear concentration range of UD to determine its ability to induce oxidative stress in Calu-3 cells in submerged culture (Figure 1A) was investigated. We observed that within 1 h of exposure, concentrations of 0.5 mg/mL and above of UD significantly increased oxidative stress in Calu-3 cells compared to the cells grown in media only. The 1 mg/mL concentration was selected for further experiments as an oxidative stress trigger while still not being cytotoxic to the cells (Figure 1C). This elicited a significantly larger increase in oxidative stress compared to 0.5 mg/mL (Figure 1A). The co-incubation study with UD and the Na-Hya and PolmonYDEFENCE/DYFESA^TM^ were conducted over 2 h (Figure 1B), with all the measurements calculated as a fold change over the 0 h time point. Menadione was used as a positive control for oxidative stress [22] and N-Acetylcysteine as a negative control. The condition with UD alone had comparable increases of oxidative stress to the positive control, validating the significance of UD in inducing oxidative stress in Calu-3 cells. After two hours of exposure, with an increased concentration of Na-Hya and PolmonYDEFENCE/DYFESA^TM^ blend, co-incubated oxidative stress was observed in cells that had the PolmonYDEFENCE/DYFESA^TM^ with an equivalent concentration of 0.5 mg/mL or above of Na-Hya when compared to control. 

Further, we conducted a cytotoxic assay using the 2 h exposure model (Figure 1C) to rule out the observed decrease in oxidative stress due to the percentage of viable cells. Our data showed that the sodium hyaluronate present in the PolmonYDEFENCE/DYFESA^TM^ could exert a protective effect against the UD and consequently a decrease in UD-induced oxidative stress.

The protective capability of PolmonYDEFENCE/DYFESA^TM^ against oxidative stress can largely be attributed to the presence of sodium hyaluronate barrier formed and that the presence of the excipient in the formulation can help to improve the effectiveness of the barrier layer, which indirectly aids in the reduction of UD-induced oxidative stress. 

### 3.2. Effects of UD and PolmonYDEFENCE/DYFESA^TM^ Blend on the Calu-3 Epithelial Layer Permeability and Integrity

To study the effect of environmental pollutants on the lung epithelium, the Calu-3 cells were grown in the ALI culture model [23]. This method of cell culture is ideal for modelling the respiratory epithelium in vitro, as it enables the cells to differentiate and form a pseudostratified epithelium which forms tight junctions and secretes mucus, with vastly different physiology compared to cells grown in a monolayer in submerged culture. Using this model, a series of experiments to test the Calu-3 epithelium integrity and permeability in response to UD were performed. HPFP, a model propellant previously shown to not affect cell integrity [19], was used to uniformly deposit the UD onto the epithelium. No significant changes in the permeability coefficient (Figure 2A) and mucus secretion (Appendix A) between cells in media vs cells with HPFP or UD were observed. The TEER experiments showed significant increases in the electrical resistance with 250 µg of UD compared to all other conditions (Figure 2A). As electrical resistance is a measure of epithelial integrity, we concluded that the exposure to UD did not have any negative effects on the tight junctions. Additionally, the Calu-3 epithelial layers were exposed to PolmonYDEFENCE/DYFESA^TM^ blend over a time course of 0 to 24 h (Figure 2B). No changes in the permeability of the layers were observed up to 6h. A small increase in permeability was observed at 8 and 24 h; however, the vehicle control 24 h epithelium showed significantly higher permeability compared to the epithelium that had the PolmonYDEFENCE/DYFESA^TM^ blend deposited on top, which indicates that the blend barrier layer was able to provide protection even after 24 h. No significant changes in mucus secretion were observed when comparing Calu-3 epithelium with and without the PolmonYDEFENCE/DYFESA^TM^ for 24 h (Appendix A), showing that a barrier layer formed can provide protection.

### 3.3. PolmonYDEFENCE/DYFESA^TM^ Blend as a Protective Barrier against Exposure to UD-Induced Inflammation 

Inflammation of the airways is a well-documented response to the inhalation of environmental pollutants and a cause of the progression of acute and chronic lung disorders [24]. During inflammation, pro-inflammatory cytokines (interleukins) and inflammatory cells accumulate and activate to induce the cytokine storm, a marker and driver of inflammation and inflammatory disease [25]. Specifically, the detection of secreted interleukin 6 (IL-6) and interleukin 8 (IL-8) is considered the gold standard method for the detection of inflammation in vitro.

The focus of the present study is to investigate the potential of the PolmonYDEFENCE/DYFESA^TM^ to provide mechanical barrier protection by reducing the contact of the urban dust with the cells layer, consequently decreasing inflammation.

Exposure to 50 µg and above amounts of UD on Calu-3 epithelial cells for 24 h resulted in significant increases in both IL-6 and IL-8 (Figure 3A). To determine if PolmonYDEFENCE/DYFESA^TM^ blend exhibited any protective abilities against UD-induced inflammation, 250 μg of UD was used in combination with Na-Hya or PolmonYDEFENCE/DYFESA^TM^ due to the large increase compared to vehicle control (HPFP) in both interleukins (Figure 3B). It was observed that 12.5 µg and 25 µg of Na-Hya alone were able to significantly decrease IL-6 production triggered by UD, while all concentrations tested of PolmonYDEFENCE/DYFESA^TM^ blend significantly decreased IL-6 secretion induced by UD.

Further, 25 µg of Na-Hya with its equivalent concentration of PolmonYDEFENCE/DYFESA^TM^ blend showed that the blend significantly decreases UD-induced IL-6 secretion compared to using the Na-Hya alone. With the presence of the excipient, the Na-Hya powder is agglomerating less with itself and dispersing more evenly by agglomerating with the mannitol (Appendix A). Thus, the better dispersion of the Na-Hya on top of the epithelium when in formulation with the excipient promotes the formation of a more even and stable barrier layer formed by the high hygroscopicity nature of Na-Hya, protecting the epithelium from direct contact with the UD.

While no significant changes in IL-8 secretion were detected using either Na-Hya or PolmonYDEFENCE/DYFESA^TM^. It is interesting to note that the increases in IL-6 secretion due to UD exposure were markedly higher than the increase in IL-8 secretion, indicating that in our model, the IL-6 pathway is the predominant pathway for inflammation-induced by UD. The presence of the barrier formed by the PolmonYDEFENCE/DYFESA^TM^ blend was able to give protection against the UD, reducing the inflammation caused by the direct contact of UD with the Calu-3 epithelial layers (Figure 3C). This effect is very positive in airway disease therapy, as IL-6 is heavily implicated in the pathogenesis of inflammatory pulmonary diseases such as asthma and COPD [26].

### 3.4. PolmonYDEFENCE/DYFESA^TM^ Blend Has a Positive Effect on Epithelial Layer Wound Healing 

Epithelial injury plays a pivotal role in the pathophysiology and progression of pulmonary epithelial diseases [27]. The airway epithelium is constantly exposed to environmental agents that may generate mechanical injury from simply breathing in pollutants, resulting in the imbalance of lung homeostasis and disruption of the epithelial barrier integrity [28]. Thus, to assess if the barrier protection mechanism of the PolmonYDEFENCE/DYFESA^TM^ blend had any positive effects on wound healing (Figure 4), a wound scratch test was performed on Calu-3 cells grown in the ALI configuration. Using time-lapse microscopy imaging, the size of the wounds (Figure 4A, lighter area marked with *) after being exposed or not to the blend, were compared. Results showed that the wound that had PolmonYDEFENCE/DYFESA^TM^ blend deposited on the surface was able to completely ‘heal’ by 19 h post wound, while HPFP and Na-Hya only ‘treated’ wounds did not. Additionally, calculating the percentage size of the wound area across multiple time points compared to time 0 (Figure 4B), a significantly greater reduction in the size of the wound with the epithelial layers that had PolmonYDEFENCE/DYFESA^TM^ blend compared to control as early as 3 h post-deposition was observed. Some significant reduction of the wound size between Na-Hya and PolmonYDEFENCE/DYFESA^TM^ blend at 5 h post-deposition was also observed.

The observed increased rate of wound healing on the Calu-3 epithelial cells that had the PolmonYDEFENCE/DYFESA^TM^ blend links well with the idea that the healing of the epithelial barrier could be aided by a therapeutic agent acting as a ‘physical barrier. Hya is a component of the extracellular matrix and is known to be involved in the wound healing process [29]. Nyman et al. demonstrated that Hya accelerates re-epithelialisation in incisional wounds [30]. While mannitol itself is not as well studied in the literature for wound healing, it can stabilise Hya as an excipient which may explain why the PolmonYDEFENCE/DYFESA^TM^ blend is much more effective than the Na-Hya alone in wound healing.

### 3.5. PolmonYDEFENCE/DYFESA Deposition

The positive in vitro biochemical effects of the PolmonYDEFENCE/DYFESA^TM^ blend open multiple avenues for its use as a protective therapy. Therefore, it is essential to determine in what areas of the respiratory system the sodium hyaluronate DPI formulation is deposited using the PillHaler^®^ DPI device. Using ACI with and without the modified plate [30], the deposition profile of the blend administered by the PillHaler^®^ Device was determined. No significant differences were detected between standard and modified plates. The majority of the PolmonYDEFENCE/DYFESA^TM^ was found deposited in the throat with 77% and 74% in standard and modified ACI, respectively. The amount of sodium hyaluronate deposited in S0 was 17% and 12% and in S1 was 5% and 4%, in standard and modified ACI, respectively, confirming that the target of the product is the upper respiratory tract up to the primary bronchi. For this reason, for the next stage of the study, the modified ACI plate (with the Calu-3 cells inserts) replaced Stage 0 of the ACI (stage S0–S1 cut-off diameter 6.5–4.4 µm).

## 4. Conclusions

This study showed a comprehensive assessment of the potential protective effect of PolmonYDEFENCE/DYFESA^TM^ product, based on an inhaled formulation of sodium hyaluronate (key ingredient) and mannitol (excipient), delivered as DPI using the PillHaler^®^ DPI device. It was observed that most of the blend was deposited in the ACI throat up to the primary stages (S0 and S1) without deposition in the lower stages of the ACI, confirming that the target of the product is the upper respiratory tract up to the primary bronchi without any deposition in the lower stages of the impactor. This aligns with its intended purpose since it could be used as a localised physical barrier for the upper respiratory tract to protect against external harmful pollutants, including potentially viruses and bacteria. Furthermore, it was demonstrated that the presence of the barrier prevents the direct contact of UD with the cells, reducing in this way the inflammatory response caused by UD. It was also observed to significantly be more effective in wound healing compared to the main ingredient alone.

While in the solubilised form, the formulation could give indirect protection against the UD and reduce levels of oxidative stress in the short term (2 h) without exhibiting any cytotoxic effects. This protection can be attributed to the viscous solution formed by sodium hyaluronate present in the product can bring to reduce the deposition of the UD on the cells and consequently guarantee its protection.

Collectively, our data demonstrate the exciting prospects of using the PolmonYDEFENCE/DYFESA^TM^ blend as a physical protective barrier against environmental pollutants. The formulation demonstrates significant potential to be used as a protective and preventative mechanism to protect the upper respiratory system in highly polluted environments or instances when an extra layer of protection for the respiratory system is desired by the user.

## Figures and Tables

**Figure 1 pharmaceutics-14-01323-f001:**
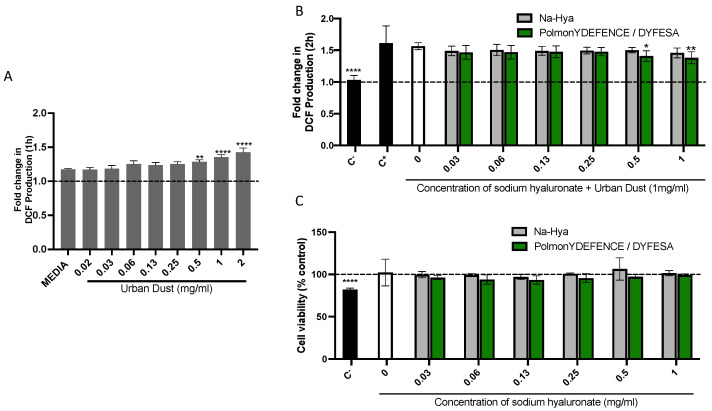
The induction of (**A**) oxidative stress induced by environmental pollutant (urban dust) in Calu-3 cells grown in submerged culture. Statistical significance was calculated by comparing all conditions to media. (**B**) Co–incubation study at 2 h exposure to urban dust and a concentration range of Na-Hya (light grey) and PolmonYDEFENCE/DYFESA^TM^ (green). Statistical significance was calculated by comparing urban dust only to every other condition. (**C**) Cell viability assay using the 2 h exposure of Calu-3 cells to the compounds. An ordinary One-Way ANOVA with Sidak’s multiple comparisons test was used (* *p* < 0.033; ** *p* < 0.0021; **** *p* < 0.0001).

**Figure 2 pharmaceutics-14-01323-f002:**
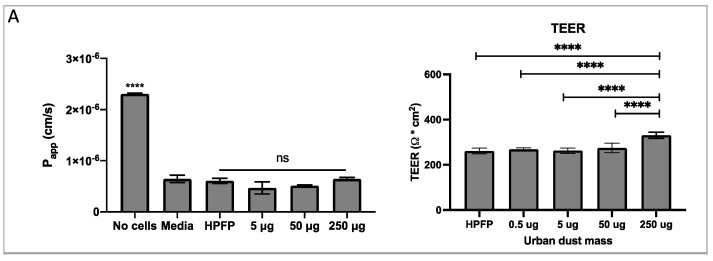
The measure of membrane integrity and permeability to test tight junction formation between cells +/− urban dust. (**A**) Permeability coefficient and TEER measurements for epithelial electrical resistance in response to varying concentrations of urban dust (**B**) Permeability coefficient and TEER measurements for epithelial electrical resistance of the epithelium exposed to PolmonYDEFENCE/DYFESA over a 24 h time course. Statistical significance was calculated using Statistical significance was calculated using one–way ANOVA with Dunnett’s multiple comparisons test (ns = no statistically significant change, # *p* < 0.033; ## *p* < 0.0021; #### *p* < 0.0001, **** *p* < 0.0001) and student *t*-test (* *p* < 0.05).

**Figure 3 pharmaceutics-14-01323-f003:**
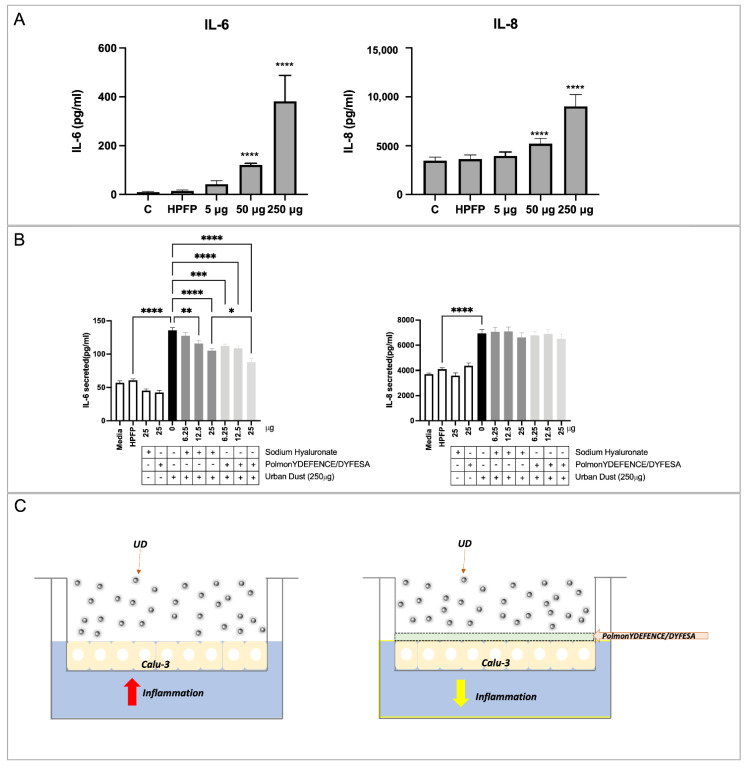
ELISA assays detecting interleukins 6 and 8 in Calu-3 cells in ALI culture using (**A**) increased concentrations of urban dust. Statistical significance is determined by comparing all conditions to vehicle control (HPFP), (**B**) using the main ingredients alone or in combination with urban dust (250 µg), statistical significance is determined between the comparison of HPFP to Media, Na–Hya only, PolmonYDEFENCE/DYFESA only, and urban dust only. Further comparisons were made between urban dust only compared to all other conditions with urban dust and Na–Hya compared to PolmonYDEFENCE/DYFESA, both with urban dust. An ordinary one–way ANOVA with Sidak’s multiple comparisons test was used (* *p* < 0.033; ** *p* < 0.0021; *** *p* < 0.0002; **** *p* < 0.0001). (**C**) Visual representation of the barrier–forming mechanism of action of PolmonYDEFENCE/DYFESA in reducing urban dust–induced epithelial inflammation.

**Figure 4 pharmaceutics-14-01323-f004:**
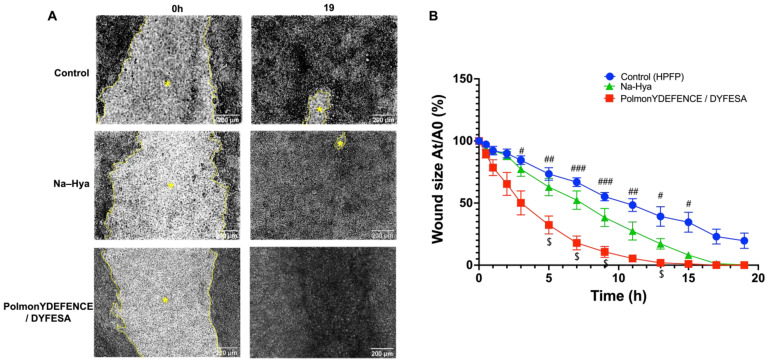
Wound healing study for PolmonYDEFENCE/DYFESA on Calu-3 epithelium in Transwells. (**A**) Time-lapse microscopy images comparing wounds at 0 h and 19 h with the brighter area outlined in yellow (*) representing the wound. (**B**) Graphical representation of the percentage of wound size of Area nt/Area 0t across a time course of 19 h to represent wound closure. Statistical significance was calculated using a two-way ANOVA with Tukey’s multiple comparisons test. Where the symbol (#) denotes significant differences between Control and PolmonYDEFENCE /DYFESA^TM^ (# *p* < 0.033; ## *p* < 0.0021; ### *p* < 0.0002) and the symbol ($) denotes significant differences between Sodium hyaluronate and PolmonYDEFENCE /DYFESA^TM^ ($ *p* < 0.033).

## Data Availability

Not applicable.

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
