# Peer review of "Protective Abilities of an Inhaled DPI Formulation Based on Sodium Hyaluronate against Environmental Hazards Targeting the Upper Respiratory Tract"

_pharmaceutics, 2022, doi:10.3390/pharmaceutics14071323_

Round 1

Reviewer 1 Report

This manuscript reported PolmonYDEFENCE / DYFESATM could protect lung epithelial cells from urban dust, or to say, environmental pollutants. The authors also indicated this protective ability was because of anti-inflammatory effects. Also, it was demonstrated that PolmonYDEFENCE / DYFESATM  was helpful to wound healing. Overall, this is an interesting paper exploring more potentials of an industrial product. Whereas it could be further improved to make it clearer what the impact or significance of this research is. And more supporting data are expected to make the conclusion more solid. Below are several minor suggestions that the authors might consider to refine the work.

  1. A flow chart or scheme of the whole research would be helpful to make it easier to understand and follow.
  2. In Figure 1a and 1b, the largest increment was smaller than 1.5 folds, which could be barely seen as a big difference.
  3. Besides IL-6 and IL-8, other representative inflammatory cytokines such as IL-1, TNF-α and IFN-γ should be assayed.
  4. Figure 3B IL-8 chart needs statistical analysis.
  5. It is not entirely clear from the manuscript how the conclusion of this study would be helpful to what group of people on what occasions in real life.

Author Response

Response to Reviewer 1 Comments

Point 1: A flow chart or scheme of the whole research would be helpful to make it easier to understand and follow.

Response 1: We have added the following figure as Supplementary Figure 1

In the introduction section page 2, line 81 we added “(Supplementary figure 1)”. The following supplementary figures have been updated to reflect the addition, page 7, line 291, page 7, line 303 and page 8, line 338.  

In the Supplementary Materials section on page 11. line 443 in the manuscript the following line have been added “Supplementary figure 1: Schematic representation of the scope of the research.”

Point 2: In Figure 1a and 1b, the largest increment was smaller than 1.5 folds, which could be barely seen as a big difference.

Response 2: We have updated the manuscript to reflect this observation by rewording the sentence “a slight but statistically significant decrease in oxidative stress was observed in cells that had the PolmonYDEFENCE / DYFESATM…” on page 6, line 260 in manuscript.

Point 3: Besides IL-6 and IL-8, other representative inflammatory cytokines such as IL-1, TNF-α and IFN-γ should be assayed.

Response 3: While a comprehensive examination of inflammatory biomarkers would be fascinating to study, the main focus of the paper is to assess the protective abilities of the barrier layer against urban dust.  IL6 and IL8 is considered the gold standard measure for inflammation (1) and sufficient for the scope of this paper. Further IL6 is considered the key cytokine for the pathogenesis of inflammatory pulmonary disease and the target of many therapeutic treatments (2) which is distinctly relevant to our study. 

  1. de Moraes MR, da Costa AC, Corrêa Kde S, Junqueira-Kipnis AP, Rabahi MF. Interleukin-6 and interleukin-8 blood levels' poor association with the severity and clinical profile of ex-smokers with COPD. Int J Chron Obstruct Pulmon Dis. 2014 Jul 29;9:735-43. doi: 10.2147/COPD.S64135. PMID: 25114519; PMCID: PMC4122580.
  2. Rincon M, Irvin CG. Role of IL-6 in asthma and other inflammatory pulmonary diseases. Int J Biol Sci. 2012;8(9):1281-90. doi: 10.7150/ijbs.4874. Epub 2012 Oct 25. PMID: 23136556; PMCID: PMC3491451.

Point 4: Figure 3B IL-8 chart needs statistical analysis.

Response 4: Figure 3B has been updated with a more comprehensive statistical analysis and the figure legend edited to reflect the extra analysis. Only statistically significant results have been annotated with an asterisk. Please see below and on page 9, lines 351-361 in manuscript.

Figure 3. ELISA assays detecting interleukins 6 and 8 in Calu-3 cells in ALI culture using (A) increased concentrations of urban dust. Statistical significance is determined comparing all conditions to vehicle control (HPFP), (B) using the main ingredients alone or in combination with urban dust (250 µg), statistical significance is determined between the comparison of HPFP to Media, Na-Hya only, PolmonYDEFENCE / DYFESA only, and urban dust only. Further comparisons were made between urban dust only compared to all other conditions with urban dust and Na-Hya compared to PolmonYDEFENCE / DYFESA both with urban dust. An ordinary One-Way ANOVA with Sidak’s multiple comparisons test was used (*p<0.033; **p<0.0021; ***p<0.0002; ****p<0.0001). (C) Visual representation of the barrier forming mechanism of action of PolmonYDEFENCE / DYFESA in reducing urban dust induced epithelial inflammation.

Point 5: It is not entirely clear from the manuscript how the conclusion of this study would be helpful to what group of people on what occasions in real life.

Response 5:

The following sentence has been added to the introduction section page 2, lines 50 in the manuscript.

“which is produced in urbanised areas because of environmental pollution.”

The following sentence has been added to the conclusions section to highlight the prospective benefits in real life on page 11, lines 434-436 in the manuscript.

“The formulation demonstrates significant potential to be used as a protective and preventative mechanism for the upper respiratory system in highly polluted environments or when an extra layer of protection for the respiratory system is desired by the user.”  

Reviewer 2 Report

In this article, they demonstrated the exciting prospects of using the PolmonYDE-FENCE/DYFESATM blend as a physical protective barrier against environmental pollutants. But, this article can be accept after major revision. The questions are as follows:

1.      There are many writing errors, such as the subscript of CO2, the superscript of cm2, etc.

2.      Scale not seen in Fig 4.

3.      Some refs should be revised, e.g.1, 3, 4, 7, 23, 29;

Author Response

Point 1: There are many writing errors, such as the subscript of CO2, the superscript of cm2, etc. 

Response 1: Manuscript has been reviewed and errors have been rectified.  

Point 2: Scale not seen in Fig 4

Response 2: Scales has been added to Figure 4

Point 3: Some refs should be revised, e.g.1, 3, 4, 7, 23, 29;

Response 3: The mentioned references have been checked for its accuracy and relevance.

Reviewer 3 Report

The paper, entitled "Protective Abilities of an Inhaled DPI Formulation Based on Sodium Hyaluronate Against Environmental Hazards Targeting the Upper Respiratory Tract" aims to  assessed the ability of an inhaled dry powder (DPI) medical device product (PolmonYDEFENCE/DYFESATM), using a formulation of sodium hyaluronate (Na-Hya) as the key ingredient as a defensive barrier to protect the upper respiratory tract. Although studies like this are important, it is not clear the importance. How do you compare with published data? How can be translated? What is the added value?

Author Response

Point 1: Although studies like this are important, it is not clear the importance. How do you compare with published data? How can be translated? What is the added value?

Response 1:

The following sentence has been added to the introduction section page 2, lines 50 in the manuscript.

“which is produced in urbanised areas because of environmental pollution.”

The following sentence has been added to the conclusions section to highlight the prospective benefits in real life on page 11, lines 434-436 in the manuscript.

“The formulation demonstrates significant potential to be used as a protective and preventative mechanism for the upper respiratory system in highly polluted environments or when an extra layer of protection for the respiratory system is desired by the user.”  

Round 2

Reviewer 3 Report

it is acceptable for publication.